# Know your enemy: Application of ATR-FTIR spectroscopy to invasive species control

**Claire Anne Holden** [1] *, **John Paul Bailey**[2], **Jane Elizabeth Taylor**[1], **Frank Martin**[3], **Paul Beckett**[4], **Martin McAinsh**[1]

1 Lancaster Environment Centre, Lancaster University, Lancaster, United Kingdom, 2 Department of Genetics and Genome Biology, Leicester University, Leicester, United Kingdom, 3 Biocel Ltd, Hull, United Kingdom, 4 Phlorum Ltd, Brighton, United Kingdom

* c.holden6@lancaster.ac.uk

**Data Availability Statement:** The datasets generated and analysed during the current study are available in a supplementary folder.

**Funding:** CAH is a member of the Centre for Global Eco-Innovation that is funded by the European

## Abstract

1. Extreme weather and globalisation leave our climate vulnerable to invasion by alien species, which have negative impacts on the economy, biodiversity, and ecosystem services. Rapid and accurate identification is key to the control of invasive alien species. However, visually similar species hinder conservation efforts, for example hybrids within the Japanese Knotweed complex.

2. We applied the novel method of ATR-FTIR spectroscopy combined with chemometrics (mathematics applied to chemical data) to historic herbarium samples, taking 1580 spectra in total. Samples included five species from within the interbreeding Japanese Knotweed complex (including three varieties of Japanese Knotweed), six hybrids and five species from the wider Polygonaceae family. Spectral data from herbarium specimens were analysed with several chemometric techniques: support vector machines (SVM) for differentiation between plant types, supported by ploidy levels; principal component analysis loadings and spectral biomarkers to explore differences between the highly invasive *Reynoutria japonica* var. *japonica* and its non-invasive counterpart *Reynoutria japonica* var. *compacta*; hierarchical cluster analysis (HCA) to investigate the relationship between plants within the Polygonaceae family, of the *Fallopia*, *Reynoutria*, *Rumex* and *Fagopyrum* genera.

3. ATR-FTIR spectroscopy coupled with SVM successfully differentiated between plant type, leaf surface and geographical location, even in herbarium samples of varying age. Differences between *Reynoutria japonica* var. *japonica* and *Reynoutria japonica* var. *compacta* included the presence of two polysaccharides, glucomannan and xyloglucan, at higher concentrations in *Reynoutria japonica* var. *japonica* than *Reynoutria japonica* var. *compacta*. HCA analysis indicated that potential genetic linkages are sometimes masked by environmental factors; an effect that can either be reduced or encouraged by altering the input parameters. Entering the absorbance values for key wavenumbers, previously highlighted by principal component analysis loadings, favours linkages in the resultant HCA dendrogram corresponding to expected genetic relationships, whilst environmental associations are encouraged using the spectral fingerprint region.

4. The ability to distinguish between closely related interbreeding species and hybrids, based on their spectral signature, raises the possibility of using this approach for determining the

Union Regional Development Fund and mediates the collaboration between Lancaster University and Phlorum Ltd.

**Competing interests:** A commercial funder, Phlorum Ltd, provided consumables funding for CAH's studentship, access to which was mediated by the Centre for Global Eco-Innovation. This does not alter our adherence to PLOS ONE policies on sharing data and materials.

origin of Japanese knotweed infestations in legal cases where the clonal nature of plants currently makes this difficult and for the targeted control of species and hybrids. These techniques also provide a new method for supporting biogeographical studies.

## Introduction

Invasive alien species (IAS), such as Japanese Knotweed, detrimentally impact the economy [1], ecosystem services [2], and native flora [3]. The impacts of IAS are set to worsen as an increasing human population heightens the demand for healthy crops [4], whilst globalisation [5] and extreme weather events [6] create further opportunities for introduction and spread of invasives. Accurate identification is the first step towards management of IAS. While many countries aim to intercept their introduction at border crossings, a lack of taxonomic experts or a world-wide comprehensive approach create barriers to identification [7].

Within the Japanese Knotweed complex, also known as *sensu lato (s.l.)*, misidentification has led to an underestimation of the prevalence of hybridisation [8,9], and similar morphology has complicated management strategies [10]. Hybridisation is a strategy employed by IAS to overcome a genetic bottleneck [11]. It results in 'heterosis', the production of offspring with increased 'hybrid vigour' [12]. Hybrid descendants may have improved traits relative to their parents such as invasiveness [11], growth rate, reproductive success and yield [13], genetic variance [14], and stress tolerance e.g. to herbicides [15] and cold [16].

The vigorous hybrid Bohemian Knotweed (*Reynoutria × bohemica)* has advantages over its maternal parent *Reynoutria japonica* var. *japonica*, including the ability to produce viable seed without the need for cross-breeding [17]. Despite its increased invasiveness it has not been recognised on the United States Department of Agriculture (USDA), Natural Resources Conservation Service (NRCS) Plants Database where it is still listed as "Absent/Unreported" in the United States of America [18]. Similarly overlooked due to morphological variation [17] is the dwarf variant *Reynoutria japonica* var. *compacta*. Misidentification is a particular concern in hybrids where viable seeds are produced because glyphosate, the main herbicide used to treat Japanese Knotweed, is applied post-flowering to increase herbicide allocation to rhizomes [19]. However, an increasing prevalence of Bohemian Knotweed [8], and the occurrence of stands (clumps) of seeding Japanese Knotweed *s.l.* [20], means late-season herbicide application may not be an appropriate 'cure-all' treatment program. Correct plant identification is therefore essential for the design of effective and stand-specific treatment programs.

Accurate identification is also important to the biocontrol of Japanese Knotweed. Two strains of psyllid currently under consideration as biocontrol agents exhibit differential development on different plant types; the northern Hokkaido biotype favours Giant Knotweed, whilst the southern Kyushu biotype prefers Japanese Knotweed and Bohemian Knotweed [21]. Consequently, accurate biogeographical information is required when sourcing biocontrol agents to ensure that introduced target species are matched with an agent from the same geographical origin. For Japanese Knotweed *s.l.*, the phylogenetic and biogeographic relationships between plants has been determined through comparison of genetic diversity levels both within, and between, introduced and native ranges [22–24]. Phylogenetic studies using current genetic methods, can require specialist techniques and knowledge [25]. Whereas the development of rapid techniques which require minimal sample preparation that can accurately distinguish between morphologically similar species in the field, complementing traditional

phylogenetic approaches, could improve the speed and accuracy of plant identification and support effective management strategies for this IAS.

Attenuated total reflection Fourier transform infrared (ATR-FTIR) spectroscopy allows the rapid, marker-free, non-destructive analysis of biological samples [26]. This technique is being increasingly used in plant science. Applications include differentiation of plants and pollen from different growing regions [27–29]; phylogenetic studies [30,31], response to abiotic factors such as soil fertility [32], heavy metals [33,34], water and temperature stress [35], nutrient deficiency and uptake [36,37]; as well as monitoring health and development [38,39] and infection [40]. Therefore ATR-FTIR spectroscopy appears well suited to fulfil a role in the identification and management of IAS.

ATR-FTIR spectroscopy measures the absorption of infrared light by a sample at specific quantifiable wavenumbers. Energy from absorbed light (4000–400 cm$^{-1}$ wavenumbers or 2.5–25 μm wavelengths) is transformed into vibrational energy through induction of atomic displacement and dipole moment changes [26]. Patterns of absorption are acquired as spectra comprising complex multivariate data that require chemometrics to derive subtle differences in sample composition. Available mathematical techniques include principal component analysis (PCA) and linear discriminant analysis (LDA), support vector machine (SVM), Naïve Bayes, and artificial neural networks (ANN) [26]. Biological molecules preferentially absorb light of wavenumbers 1800–900 cm$^{-1}$, a range known as the 'fingerprint region', which includes important biological absorptions due to lipids, proteins, carbohydrates, nucleic acids and protein phosphorylation (see [26]). Databases are available with catalogued definitions for characteristic peak frequencies (e.g. [41]). For example, absorptions have been linked to biologically significant compounds including glucomannan [42], xyloglucan [43], succinate [44], and pectin [45]. However, the process from chemometric biomarker identification to physical biomolecular extraction is the subject of ongoing research focused on calibrating concentrations derived from biological spectra [46], consolidating the expanding database of key wavenumber changes and associated molecular definitions [41], and trialling new biological applications [37,39,40].

This study aims to develop a tool to support management strategies through clarification of IAS species assignment, population dynamics, and biogeography. Spectral data were analysed using a combination of mathematical techniques: principal component analysis (PCA) to assess the natural variation; the classifier support vector machines (SVM), a supervised technique, to allow identification of closely related interbreeding species and hybrids, supported by ploidy levels; PCA-LDA and biomarkers to elucidate the biochemical differences between invasive and non-invasive varieties; and hierarchical cluster analysis (HCA) to explore species phylogeny.

## Materials and methods

### Herbarium samples

Samples were obtained from the University of Leicester herbarium (LTR), previously collected between 1935–2000, see S1 Table for detailed sample information. Leaves were air-dried at the time of collection and subsequently stored in cardboard folders within purpose-built cupboards to protect them from light exposure. Sample types included five interbreeding species, six hybrids, and five more distantly related 'out species' which were included in the analysis for the study of phylogeny, see Table 1.

Species assignments were confirmed at the time of collection based on chromosome numbers (John Bailey, personal communication). For large leaves a four cm$^2$ square was cut out of each leaf between the second and third veins from the bottom left corner of each sample, to

**Table 1. Species information for samples within the Polygonaceae family.**

| Latin Name | Contextual information |
|---|---|
| *Reynoutria japonica* **var.** *japonica* | • 'true' Japanese Knotweed<br>• Western accessions are octoploid, 4n = 8x = 88<br>• Giant tetraploid plants are known in Japan, 2n = 4x = 44 |
| *Reynoutria japonica* **var.** *compacta* | • 'dwarf' Japanese Knotweed<br>• exclusively tetraploid, 2n = 4x = 44 |
| *Reynoutria japonica* **var.** *uzenensis* | 'hairy' Japanese Knotweed |
| *Fallopia baldschuanica* | Russian vine |
| *Reynoutria sachalinensis* | • Giant Knotweed<br>• predominantly tetraploid, 2n = 4 x = 44 |
| *Reynoutria x bohemica*<br>**OR**<br>*Reynoutria japonica* **var.** *japonica* **x** *sachalinensis* | • Bohemian Knotweed (the most common hybrid)<br>• 'True' Japanese Knotweed crossed with Giant Knotweed<br>• predominately hexaploid, 2n = 6x = 66 |
| *Reynoutria sachalinensis* **x** *Fallopia baldschuanica* | Giant Knotweed crossed with Russian Vine |
| *Reynoutria japonica* **var.** *japonica* **x** *Fallopia baldschuanica* | 'True' Japanese Knotweed crossed with Russian Vine |
| *Fallopia japonica* **var.** *compacta* **x** *baldschuanica* | Dwarf variety of Japanese Knotweed crossed with Russian Vine |
| *Reynoutria japonica* **var.** *compacta* **x** *Reynoutria sachalinensis* | dwarf variety of Japanese Knotweed crossed with Giant Knotweed |
| *Fagopyrum esculentum* | Buckwheat |
| *Rumex acetosella* | Sheep's Sorrel |
| *Fallopia convolvulus* | Black-bindweed |
| *Fallopia multiflora* | Tuber Fleece-flower |
| *Fallopia cilinodis* | Fringed Bindweed |

preserve the herbarium samples for future users. For small leaves where this was not possible, the whole leaf was taken.

## ATR-FTIR spectroscopy

Herbarium samples were analysed using a Tensor 27 FTIR spectrometer with a Helios ATR attachment (Bruker Optics Ltd, Coventry, UK). Ten spectra were taken from each leaf surface, resulting in twenty spectra per sample, 1580 spectra in total. A camera attachment was used to locate the area of interest and ensure an even spread of spectra across each surface for minimisation of bias. The ATR diamond crystal was cleaned between measurements of each leaf surface with wipes containing isopropyl alcohol (Bruker Optics, Coventry, UK), and background spectra were taken each time to account for ambient atmospheric conditions. Leaf material was placed on a slide with the analysed side facing upward, and then raised using a moving platform to make consistent contact with the Internal Reflection Element, a diamond crystal, defined the sampling area as 250 μm x 250 μm. Spectral resolution was 8 cm$^{-1}$ with 2 times zero-filling, giving a data-spacing of 4 cm$^{-1}$ over the range 4000 to 400 cm$^{-1}$; 32 co-additions and a mirror velocity of 2.2 kHz were used for optimum signal-to-noise ratio.

## Spectral data handling and analysis

Acquired spectra were converted from OPUS format to.txt files before input to MATLAB (MathWorks, Natick, USA). Pre-processing of acquired spectra is essential for spectroscopic experiments to improve the signal-to-noise ratio, reduce spectral baseline distortions, and correct systematic variations in the absorbance intensity caused by different sample thickness [47]. Pre-processing and computational analysis of the data were performed using an in-house

developed IRootLab toolbox [48,49] and the PLS Toolbox version 7.9.3 (Eigenvector Research, Inc., Manson, USA), according to standardised protocols for analysis of biochemical spectra [26,50]. Spectra were cut at the biochemical fingerprint region (1800-900 cm$^{-1}$), Savitzky-Golay (SG) second differentiated, and vector normalised. All data were mean-centred before multivariate analysis.

The natural variation between samples was explored using the unsupervised technique, principal component analysis (PCA) [51]. For the classification of groups, PCA was followed by linear discriminant analysis (PCA-LDA) [52] and the more complex technique, support vector machines (SVM) [53]. The classification hyperplane found by SVM provided the largest margin of separation between the data clusters. This was achieved using the most common kernel function, the radial basis function (RBF) [54], that transformed the data into a different feature space during model construction. PCA-LDA was constructed using 10 principal components (PCs). The number of components of PCA-LDA and all SVM parameters were optimized by venetian blinds (10 data splits) cross-validation. Spectra were randomly divided into a training set (70%, 1106 spectra) and an external test set (30%, 474 spectra) to perform validation. For SVM parameters cost, gamma and number of support vectors see S2 Table. Increasing the cost and gamma values increases the complexity of the model. This makes the margin of separation between categories more specialised to the training data set, which results in fewer misclassifications but reduces the generalisation of the model.

The main spectral alterations were characterised with PCA loadings, for which peak maxima were identified with a peak-pick algorithm (20 cm$^{-1}$ minimum separation). These spectral biomarkers were matched with previously characterised wavenumbers to give tentative chemical assignments. Further detail on biomarkers can be gained from comparisons of band intensity in the baseline corrected spectra (see S1 Fig) and horizontal shifts in the vector normalised spectra, which indicate concentration and molecular structural alteration respectively.

Sample relationships were explored using the unsupervised pattern recognition method, hierarchical cluster analysis (HCA) and the resultant hierarchy was depicted in the form of a dendrogram (see Fig 5A–5C). Clustering was achieved using Euclidean distance as the metric of sample similarity and Ward's Method as the linkage criterion. HCA was used to analyse both the fingerprint region and the PCA loadings of the samples. Where genetic linkages were overridden by environmental factors, the wavenumbers highlighted by PCA loadings rather than the fingerprint region were used for the HCA analysis. For the fingerprint region the spectra were first averaged by sample type, then pre-processed by SG differentiation, vector normalisation, before analysis by HCA. For the HCA based on the loading data, spectra were first pre-processed using second differentiation and vector normalisation. The absorbance values wavenumbers highlighted in the PCA loadings were selected, and spectra were averaged by sample type. HCA was then performed on the loadings and a dendrogram was produced.

## Results

### Sample types can be differentiated using ATR-FTIR spectroscopy and chemometrics

ATR-FTIR spectroscopy was used to explore the relationship between sample types (i.e. species, variety, hybrid, see Table 1), and to determine whether herbarium samples remain tractable to this type of analysis decades after collection. Fig 1A and 1B shows the raw and pre-processed spectra, where the mean spectra at the fingerprint region (1800–900 cm$^{-1}$) are grouped by sample type. Major change trends in intensity can be highlighted by visual comparison of averaged second-derivative spectra [55]. These spectra show clear differences; for example, some samples have peaks which others lack, or horizontal shifts visible at certain peaks.

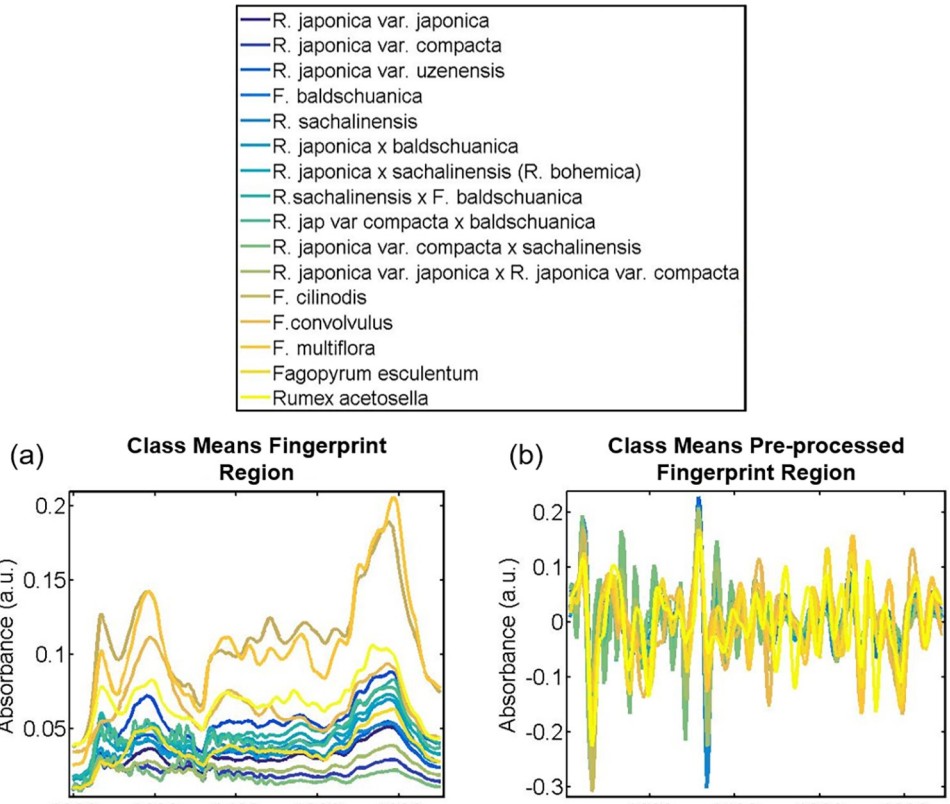

**Fig 1.** (a) Raw and (b) pre-processed class means IR-spectra for fingerprint region grouped by species. The pre-processing used for part (b) was Savitzky-Golay (SG) second differentiation followed by vector normalisation.

Fig 2 shows that differences in the spectral fingerprint region (1800–900 cm$^{-1}$) of herbarium samples were sufficient to identify between sample types using PCA, PCA-LDA and SVM analyses. Importantly, the classification of samples by a combination of ATR-FTIR spectroscopy and chemometrics was consistent with chromosome counts performed at the time of collection (John Bailey, personal communication). Fig 2A shows the PCA scores which indicate the natural variation between samples. Although some clustering of spectra can be seen, a clear separation between samples was not observed in the scores on PC1 and PC2, which indicates high similarity between spectral profiles and the requirement for supervised analysis methods.

The predictive capability of PCA-LDA, as a supervised test was however also limited, with sample types overlapping and few distinct clusters (Fig 2B). Overall, PCA-LDA gave 86% accuracy, 52.30% sensitivity and 91.51% specificity (see S3 Table for each type individually). Within this, the sensitivity scored 0% using PCA-LDA for two types: *Reynoutria japonica* var. *japonica x Reynoutria japonica* var. *compacta* and *Reynoutria japonica x baldschuanica*. In addition, of the *Reynoutria japonica* var. *japonica x Reynoutria japonica* var. *compacta* spectra, 75% were mistaken for *Reynoutria japonica x sachalinensis (Reynoutria x bohemica)*.

For the plant type *Reynoutria japonica x baldschuanica*, spectra were acquired from only two samples (a plant artificially crossed at Leicester University by Dr. John Bailey and a plant collected from Cornwall) resulting in 40% of spectra being assigned incorrectly to *Reynoutria japonica* var. *compacta x sachalinensis*, another artificial hybrid. This suggests that environmental factors can outweigh the correct genetic assignment of samples when the sample

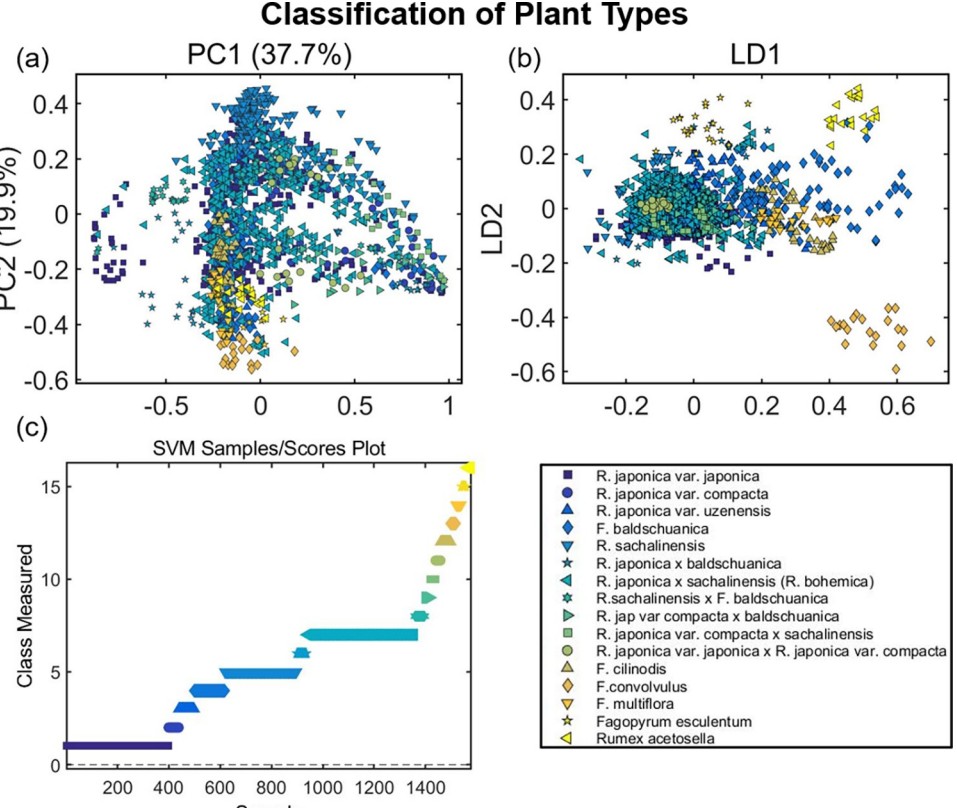

**Fig 2.** (a) PCA, (b) PCA-LDA and (c) SVM of IR-spectra taken from both leaf surfaces for fingerprint region (1800–900 cm$^{-1}$) grouped by species, for all sixteen species with both sides of leaves included. Prior to multivariate analysis, the spectal fingerprint region was pre-processed using Savitzky-Golay (SG) second differentiation followed by vector normalisation and finally mean-centring.

numbers are too low to provide sufficient training data and highlights the need for caution if using PCA-LDA as a classification method for comparison of closely related hybrids and inter-breeding species. In contrast, PCA-LDA achieved 100% specificity for the assignment of *Fallopia convolvulus*, *Fallopia multiflora*, and *Fagopyrum esculentum* using only one sample leaf each, which is likely a reflection of the distinct genetic nature, and therefore biochemical composition, of these species compared to the other plant types studied [56].

SVM achieved excellent performance in both training (100% accuracy) and test sets (99.04% accuracy), successfully differentiating plant types based on their IR spectral profile (Fig 2C). SVM gave an average of 98.25% sensitivity and 98.32% specificity (see Table 2 for each type individually). However, focusing on the eleven most closely related species, then the average specificity increases to 100% for distinguishing between interbreeding species and hybrids of Japanese Knotweed, see S4 Table. Unlike PCA-LDA, sensitivity remained high when using SVM despite a small available training set.

## ATR-FTIR spectroscopy can distinguish between leaf surfaces in samples from the herbarium

PCA was used to explore natural differences between adaxial (upper) and abaxial (lower) leaf surfaces. Clustering of spectra from upper and lower leaf surfaces can be seen in the PCA scores (S2 Fig). PCA-LDA was constructed using 10 PCs, differentiating spectra from the

**Table 2. Quality parameters (accuracy, sensitivity, and specificity) for spectral classification based on sample type of closely related species, hybrids, and varieties by SVM.**

| SVM | % Accuracy | % Sensitivity | % Specificity |
|---|---|---|---|
| *Reynoutria japonica* var. *japonica* | 98.61 | 98.49 | 98.50 |
| *Reynoutria japonica* var. *compacta* | 96.22 | 92.50 | 93.02 |
| *Reynoutria japonica* var. *uzenensis* | 100.00 | 100.00 | 100.00 |
| *Fallopia baldschuanica* | 99.13 | 98.32 | 98.35 |
| *Reynoutria sachalinensis* | 98.81 | 97.86 | 97.90 |
| *Reynoutria japonica x baldschuanica* | 97.40 | 94.87 | 95.12 |
| *Reynoutria japonica x sachalinensis (Reynoutria x bohemica)* | 98.32 | 97.15 | 97.21 |
| *Reynoutria sachalinensis x Fallopia baldschuanica* | 98.72 | 97.50 | 97.56 |
| *Reynoutria japonica* var. *compacta x baldschuanica* | 97.59 | 95.24 | 95.45 |
| *Reynoutria japonica* var. *compacta x sachalinensis* | 99.94 | 100.00 | 100.00 |
| *Reynoutria japonica* var. *japonica x Reynoutria japonica* var. *compacta* | 99.94 | 100.00 | 100.00 |
| *Fallopia cilinodis* | 100.00 | 100.00 | 100.00 |
| *Fallopia convolvulus* | 100.00 | 100.00 | 100.00 |
| *Fallopia multiflora* | 100.00 | 100.00 | 100.00 |
| *Fagopyrum esculentum* | 100.00 | 100.00 | 100.00 |
| *Rumex acetosella* | 100.00 | 100.00 | 100.00 |
| **Average** | **99.04** | **98.25** | **98.32** |

upper and lower surfaces along axis LD1 of 2D scatterplot, see S2C Fig. Classification performance of PCA-LDA was good with 74.1% accuracy, 72.3% sensitivity, and 75.9% specificity for the upper leaf surface, and 74.1% accuracy, 75.9% sensitivity, and 72.3% specificity for the lower leaf surface. SVM performed even better achieving a clear distinction between upper and lower surfaces. SVM test sets achieved an average of 98.4% accuracy, 98.4% sensitivity, and 98.4% specificity, see S2D Fig.

## Herbarium samples can be classified by their geographical origin using ATR-FTIR spectroscopy

The material analysed in this study were of Japanese Knotweed (*Reynoutria japonica* var. *japonica*) collected from the Scottish island of Shetland, and various locations across England and Japan (see S5 Table). PCA was used to explore natural differences in plants between regions clustering of spectra into samples from England, Shetland, and Japan (Fig 3A). PCA-LDA resulted in an average 87.85% accuracy, 80.19% sensitivity, and 90.20% specificity, see S5 Table. Good separation was observed in the 2D scatterplot of spectra along the axes LD1 and LD2, revealing differences between spectra from different geographical locations (Fig 3B). Spectra from English samples separated from the other two regions along the LD2 axis. Spectra from Shetlandic samples separated from the other two along the axis LD1. SVM performed the best, creating a clear distinction between the three locations (Fig 3C), and achieving 100% in accuracy, sensitivity, and specificity (Fig 3D).

## Chemometric analysis of spectral data highlights differences between dwarf and invasive knotweed varieties

Molecular differences between two varieties, *Reynoutria japonica* var. *japonica* and *Reynoutria japonica* var. *compacta* were examined using PCA-LDA analysis (Fig 4A). The latter variety is intriguingly non-invasive and rarely naturalises [57]. PCA loadings (Fig 4B) were subsequently used for the identification of biomarkers (Table 3). The PCA-LDA distributions of the two

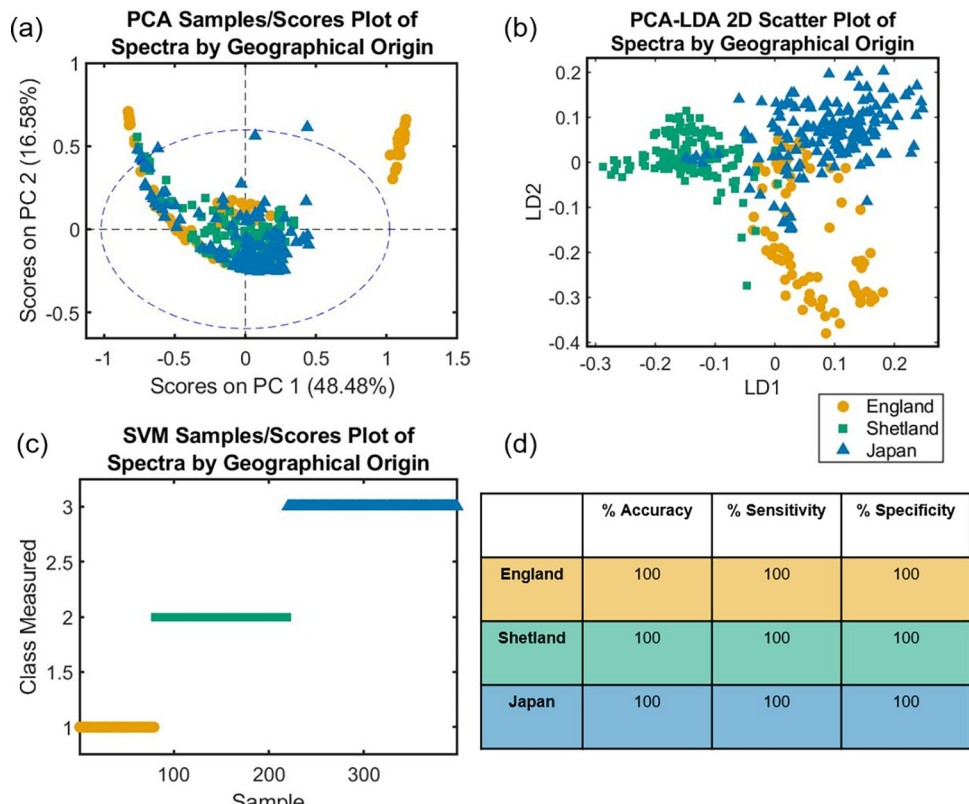

**Fig 3.** (a) PCA scores plot, (b) LDA 2D scatter plot, (c) SVM scores plot and (d) SVM classification table of fingerprint spectra (1800–900 cm$^{-1}$) grouped by geographical origin of *Reynoutira japonica* var. *japonica* samples: England (orange), Shetland (green) and Japan (blue). Prior to multivariate analysis, the spectal fingerprint region was pre-processed using Savitzky-Golay (SG) second differentiation followed by vector normalisation and finally mean-centring.

species differ in shape, with the mean clearly separated along the axis LD1 (Fig 4A). The following biomarkers were present at higher concentrations in *compacta* than *japonica*: 1744 cm$^{-1}$ (ester carbonyl group of triglycerides; [58], 1682 cm$^{-1}$ (succinic acid; [44] or the β-turns of Amide I; [59], and 1485 cm$^{-1}$ (C$_8$-H coupled with a ring vibration of guanine [41]. In contrast, the following biomarkers were present at higher concentrations in japonica than compacta: 1639 cm$^{-1}$ (Amide I [41], 1443 cm$^{-1}$ (δ(CH) of pectin) [45], 1396 cm$^{-1}$ (symmetric CH$_3$ bending of the methyl groups of proteins) [41], 1339 cm$^{-1}$ (in-plane C-O stretching vibration combined with the ring stretch of phenyl [41], 1142 cm$^{-1}$ (phosphate and oligosaccharides) [41], 1034 cm$^{-1}$ (glucomannan) [42], and 945 cm$^{-1}$ (xyloglucan) [43]. Particularly noticeable was the peak at 1034 cm$^{-1}$ corresponding to the polysaccharide, glucomannan [42];which was two-fold higher in the invasive variety. In addition, horizontal shifts indicative of structural changes were present in the following biomarkers: 1744 cm$^{-1}$ (ester carbonyl group of triglycerides) [58], 1443 cm$^{-1}$ (δ(CH) of pectin) [45], 1142 cm$^{-1}$ (phosphate and oligosaccharides) [41], 945 cm$^{-1}$ (xyloglucan) [43].

## ATR-FTIR spectroscopy offers a rapid alternative to genetic methods of phylogenetic analysis

To determine the potential of ATR-FTIR spectroscopy and multivariate analysis for phylogenetic studies, and specifically within the Polygonaceae family, the relationship between all sixteen plant types studied within this family was considered (Fig 5A–5C). The resultant linkages

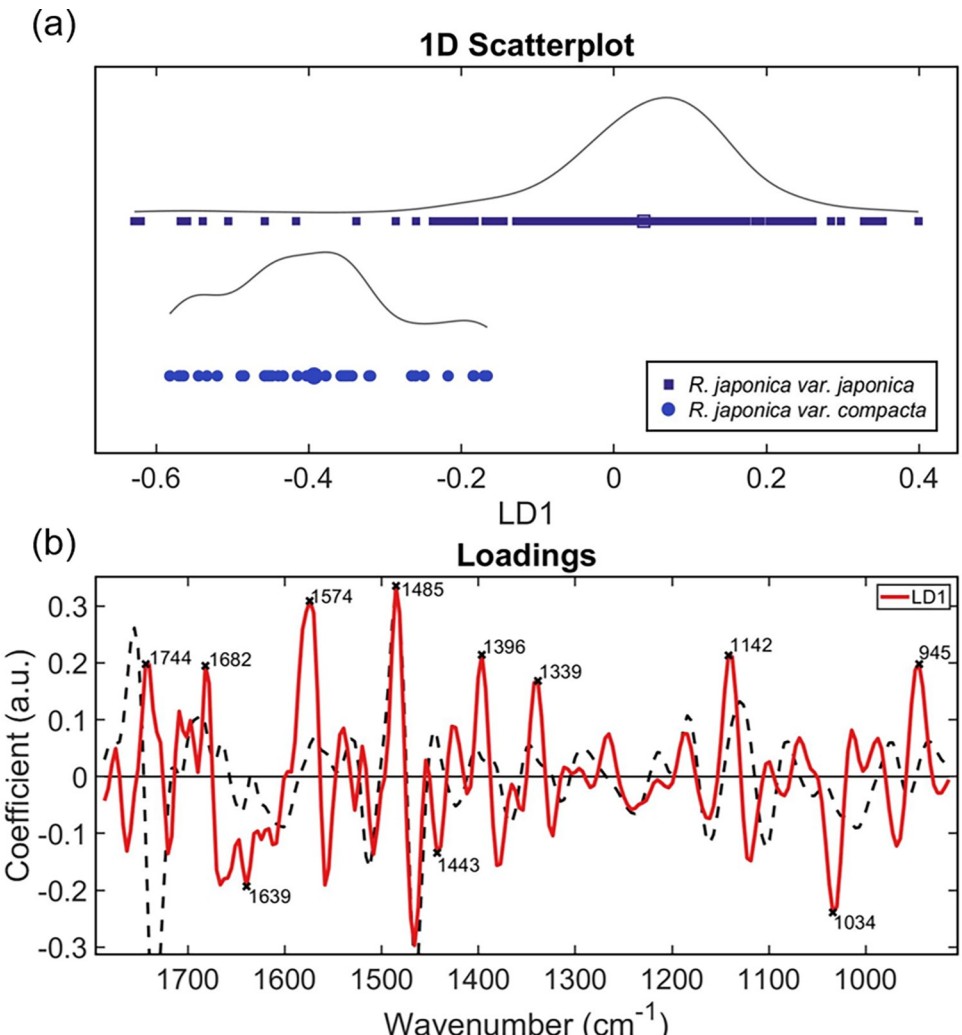

**Fig 4.** (a) PCA-LDA (b) loadings for *Reynoutria japonica* var. *japonica* vs *Reynoutria japonica* var. *compacta*. Fig 4B depicts the PCA loadings in red, and the total mean spectrum as the black dashed line, scaled to fit. Prior to multivariate analysis, the spectal fingerprint region was pre-processed using Savitzky-Golay (SG) second differentiation followed by vector normalisation and finally mean-centring.

**Table 3. Main wavenumbers responsible for class differentiation between the highly invasive *Reynoutria japonica* var. *japonica* and its more easily controllable counterpart *Reynoutria japonica* var. *compacta*, and their assigned biomarkers.**

| Wavenumber/cm | Assignment | Reference |
|---|---|---|
| 1743.65 | Ester carbonyl group C = O of triglycerides | [58] |
| 1681.93 | Succinic acid (in pure solid form) Amide I, β-turns | [44,59] |
| 1639.49 | Amide I | [41] |
| 1573.91 | C = N adenine | [41] |
| 1485.19 | $C_8$-H coupled with a ring vibration of guanine | [41] |
| 1442.75 | δ(CH) of pectin | [45] |
| 1396.46 | Symmetric $CH_3$ bending of the methyl groups of proteins | [41] |
| 1338.6 | In-plane C-O stretching vibration combined with the ring stretch of phenyl | [41] |
| 1141.86 | Phosphate and oligosaccharides; oligosaccharide C–O bond in hydroxyl group might interact with some other membrane components | [41] |
| 1033.84 | Glucomannan | [42] |
| 945.119 | Xyloglucan | [43] |

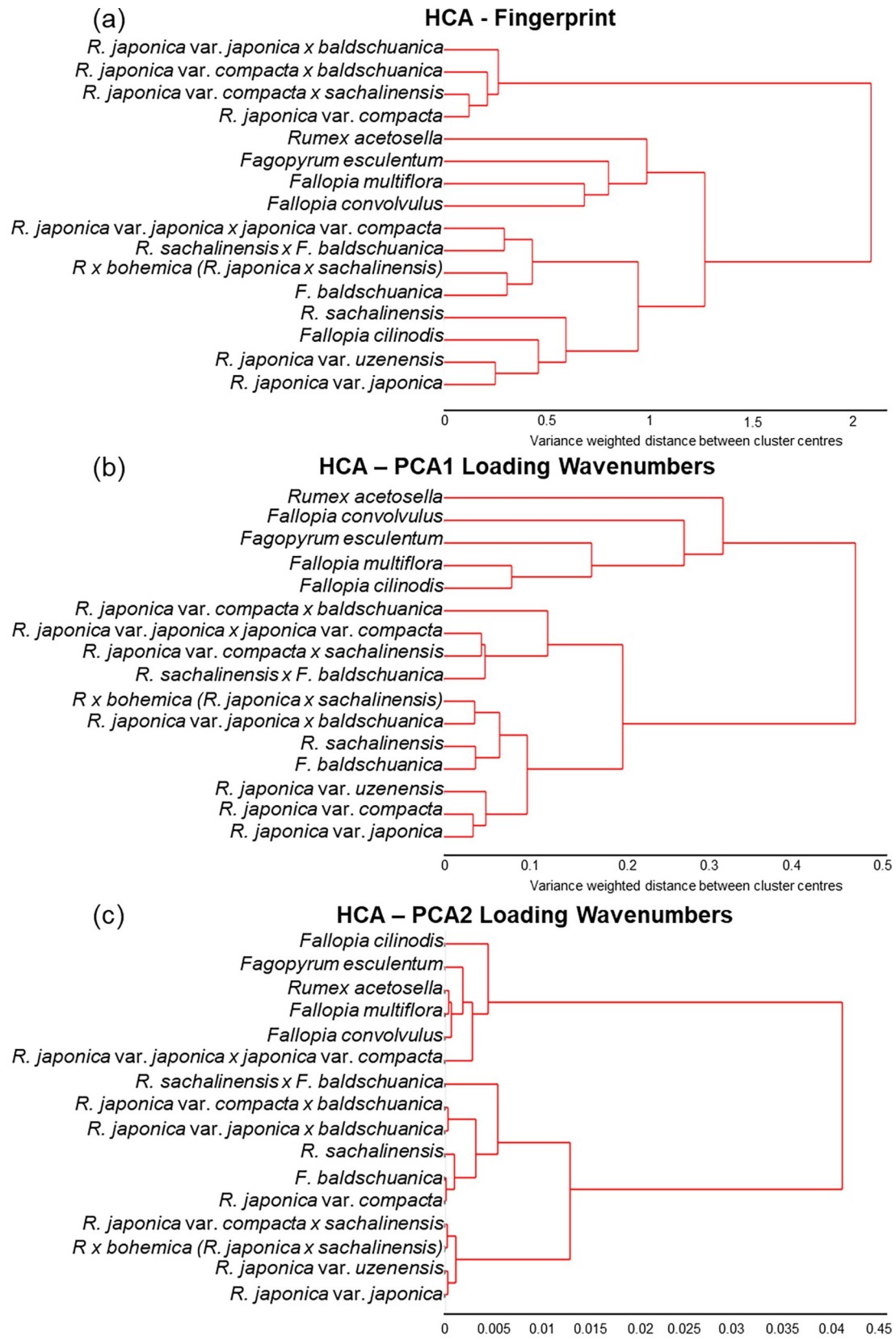

**Fig 5.** Hierarchical cluster analysis dendrogram results based on the Euclidean distance and Ward's method, comparison based on (a) spectral fingerprint region (1800–900 cm$^{-1}$), or the wavenumbers from the (b) PC1 and (c) PC2 loadings. Prior to multivariate analysis, the spectal fingerprint region in part (a) was pre-processed using Savitzky-Golay (SG) second differentiation followed by vector normalisation and finally mean-centring.

derived from HCA analysis differed depending on the kinds of spectral information compared: the fingerprint region (Fig 5A); the wavenumbers from the PC1 and PC2 loadings (Fig 5B); or for loadings graphs (S3 Fig).

Using the fingerprint region, environmental growth conditions were the dominant influence over the relationship (Fig 5A). The most distantly related group in this linkage comprised four species grown in 'captivity': *Reynoutria japonica* var. compacta grown in Cambridge Botanic Gardens, and three artificial hybrids (cultivated by hand-pollination) grown at University of Leicester (UK), *Reynoutria japonica* var. compacta x sachalinensis, *Reynoutria japonica* var. *compacta x baldschuanica* and *Fallopia japonica* var. *japonica x baldschuanica*. Within this group, the parent dwarf variety and the artificial dwarf hybrids are genetically similar, however they group further away from *Reynoutria japonica* var. *japonica* than expected. The next cluster contained four out of the five 'out species', with the more distantly related *Rumex* and *Fagopyrum* genera placing further away than the genetically closer *Fallopia* species. Surprisingly, the fifth 'out species', *Fallopia cilinodis*, was placed within the *Reynoutria* species.

In contrast, when the wavenumbers from the PC1 loadings were selected for HCA (Fig 5B), the results more closely matched the genetic relationship. Two main clusters separated the 'out-species' from Japanese Knotweed *s.l.*. Within the knotweed group, artificial greenhouse grown hybrids group together. The parental species which can interbreed with Japanese Knotweed, *Reynoutria sachalinensis* and *Fallopia baldschuanica*, group together with their *Reynoutria japonica* hybrids, including the key naturally occurring hybrid *Reynoutria x bohemica*. All three varieties of Japanese Knotweed then grouped together in a close cluster.

When HCA was performed using the wavenumbers from the PC2 loadings (Fig 5C), the order of the out species was no longer reflective of genetic relationship with *Fallopia cilinodis* grouping furthest away despite its *Fallopia* genus and the *Reynoutria. japonica* var. *japonica x Reynoutria japonica* var. *compacta* unexpectedly grouping with the more genetically distant species. However, the artificial hybrids grouped together with parent species *Fallopia baldschuanica*, *Reynoutria sachalinensis*, and *Reynoutria japonica* var. *compacta* despite being artificial and greenhouse grown. The dwarf and japonica knotweed varieties crossed with *Reynoutria sachalinensis* grouped together as genetically similar and placed closer to the *Reynoutria japonica* parent than *Reynoutria sachalinensis*.

## Discussion

### ATR-FTIR spectroscopy combined with chemometrics as a tool for IAS identification

Our results show that ATR-FTIR spectroscopy of historic herbarium samples followed by analysis with SVM is able to effectively differentiate between plant type, even closely related hybrids, achieving 99% accuracy. This result opens the possibility of applying this method to historic samples to validate the species or hybrid assignment and conclusions drawn from previous studies, which have lacked cytological confirmation [8]. This also suggests that this approach offers a solution to the misidentification and underestimation of hybridisation caused by morphological similarities within Japanese Knotweed *s.l.* [8] which has previously complicated management strategies [10]. Additionally, it raises the intriguing possibility of

being able to distinguish between, or confirm the identity and sources of, phenotypically identical knotweed populations when apportioning responsibility for the damage caused by this species to property, which is a particular issue in the UK [60].

Interestingly, both PCA-LDA and SVM successfully differentiated between leaf surfaces. The observed differences (see S2 Fig) are consistent with the presence of trichomes on the abaxial epidermis, differences which are also used to visually categorise hybrids and species within Japanese Knotweed *s.l.* [57], and the different functions performed by the two surfaces, with the upper (adaxial) surface primarily acting to conserve water and the lower (abaxial) surface more commonly involved in gas exchange [61]. Water conservation is often enhanced by a thicker waxy cuticle on adaxial leaf surfaces, comprising different epicuticular waxes: adaxial containing primary alcohols and esters, and waxes on the abaxial surface comprised of alkanes, aldehydes and secondary alcohols [62,63]. Leaf surfaces have differing responses to the environment. For example epidermal wax composition alters susceptibility to fungal pathogens [63] and differing abaxial and adaxial epidermal $Ca^{2+}$ concentrations affect stomatal guard cell sensitivity [64]. Therefore, the choice of leaf surface should be a consideration for method design if applied to monitoring of disease by fungal pathogens, for example in the context of biocontrol agents. Variations in epidermal thickness may also play a role in species classification. The infrared light used for ATR-FTIR spectroscopy only penetrates ~0.5–2 µm deep into the sample, compared with the average plant cuticle thickness of ~1–10 µm; this may allow spectral acquisition of different compounds from deeper within the leaf in species with thinner cuticles. For example, *Reynoutria sachalinensis* leaves are thinner overall with a thicker cuticular layer than those of *Reynoutria japonica*, see S4 Fig for scanning electron microscope images.

## ATR-FTIR spectroscopy captures environmental information

Spectra of *Reynoutria japonica* var. *japonica* from England, Japan and the Scottish isle of Shetland were clearly distinguished by location using the SVM algorithm (Fig 3C), achieving 100% in accuracy, sensitivity, and specificity (Fig 3D). Japanese samples were likely genetically diverse as these plants can reproduce sexually within their native range and adapt to different environmental locations [65]. However, British specimens are believed to all originate from the female clone introduced in 1850 by Philipp von Siebold [66]. Nevertheless, Shetlandic samples were distinct and effectively separated from both those from England and Japan, suggesting a strong environmental influence on spectra.

The differences detected using this approach between putatively genetically identical clonal plants from different geographical locations could be due, at least in part, to phenotypic plasticity, where one genotype can express different phenotypes [67]. Phenotypic plasticity is a result of environment-genotype interactions. This is a particularly significant mechanism for invasive plants, a high percentage of which are clonal, such as Japanese Knotweed [68]. Epigenetic modifications, somatic mutations, resource provisioning and biochemical functioning may contribute to the phenotypic plasticity allowing successful invasion of Japanese Knotweed in a diverse range of habitats [69–71], particularly as clonal plants are able to bypass the meiotic resetting of epigenetic modifications through asexual reproduction [68].

ATR-FTIR spectroscopy examines the biochemical fingerprint of the leaf surface and environmental effects are reflected within the spectra, for example, the grouping of artificial greenhouse grown hybrids together in Fig 5A–5C. Intriguingly, when spectra were grouped by individual sample rather than sample type and fingerprint regions were compared by HCA, a mixture of genetic and environmental factors appeared to influence the results. Samples group into clusters based on their geographical source or their genetic species class (see S5 Fig). This observation has potential applications in the context of biocontrol of IAS. Biocontrol agents

must be carefully selected for maximum efficacy, in the case of Japanese Knotweed taking into account the host preferences of different psyllid biotypes [21]. It is therefore important, when deciding from where to obtain biocontrol agents, to have access to combined genetic and geographical information about host plants in their native range (Pashley, 2003). In this context the weaving together of genetic and environmental information could prove an invaluable biogeographical tool.

In the present study, the fingerprint region (1800–900 cm$^{-1}$) of acquired spectra was selected for chemometric analysis because biological molecules preferentially absorb light of these wavenumbers, including important biological absorptions due to lipids, proteins, carbohydrates, nucleic acids and protein phosphorylation [26]. Isolation of this fingerprint region (1800–900 cm$^{-1}$) has achieved good results in other plant studies [29,39,40,72–74], though the high region (3700–2800 cm$^{-1}$) has also yielded valuable information in a range of applications [75–78] since it contains additional biologically relevant absorbances such as those for water (~3275 cm$^{-1}$), protein (~3132 cm$^{-1}$), fatty acids and lipids (~3005, ~2970, ~2942 and ~2855 cm$^{-1}$) [26].

## Chemometric analysis of spectral data provides insights into why the dwarf variety of knotweed is less invasive than 'true' knotweed

Unexpectedly, two compound types involved with energy production, triglycerides and succinate, were higher in concentration in the dwarf variety of knotweed than the invasive variety (Fig 4). In vegetative tissues, triacylglycerol metabolism is used as an energy source for cell division and expansion, stomatal opening, and membrane lipid remodelling [79], whilst succinate is involved in the production of ATP and acts as a signalling hub [80]. One explanation for this observation might be the differential effects of environmental conditions, time of day, or age of leaf of the sample at the time of collection [81]. The structural differences in triglycerides, as indicated by a horizontal spectral shift, is consistent with the knowledge of their construction. Triglycerides are tri-esters comprised of a glycerol bound to three fatty acid molecules, the R-groups of which can vary in chirality and composition [82]. Further work using samples from plants grown under controlled conditions is therefore required to investigate this observation before definitive conclusion can be drawn.

Conversely, pectin (a compound which strengthens the plant cell wall), oligosaccharides and two polysaccharides (glucomannan and xyloglucan) were present in higher concentrations in the invasive Japanese Knotweed variety compared with the dwarf variety (Fig 4). Interestingly, the C–O bond in the hydroxyl group responsible for the oligosaccharide wavenumber biomarker might interact with other membrane components [41], leading to the indicated structural change. Plant oligosaccharides, which include fructans and raffinose family oligosaccharides (RFOs), act as important multifunctional compounds. They can act as reserve carbohydrates, membrane stabilizers and stress tolerance mediators, play a role in osmoregulation and source–sink relationships, contribute to overall cellular reactive oxygen species (ROS) homeostasis by specific ROS scavenging, and act as phloem-mobile signalling compounds under stress [83]. In addition, the observed structural change in xyloglucan is consistent with the known species-dependent branching pattern of this molecule [84]. Xyloglucan is the most abundant hemicellulosic polysaccharide in the primary cell wall of most vascular plants, and together with cellulose gives the wall its strength. Glucomannan and xyloglucan act as storage polysaccharides in tubers and seeds. The presence of higher levels in invasive 'true' knotweed than in its dwarf counterpart raises the intriguing possibility that these polysaccharides could contribute to the comparatively rapid growth of *Reynoutria japonica* var. *japonica*.

## HCA analysis of PCA loadings offers a rapid alternative to traditional phylogenetic analysis

The power of this technique has already been shown for phylogenetic studies in some flowering plants [30] and agronomically important species such as wheat [31]. The results of the present study show potential for application to invasive species. ATR-FTIR spectroscopy combined with HCA analysis shows potential as a complementary technique alongside genetic methods to explore phylogeny and biogeographic relationships, without prior sequence knowledge. The HCA dendrogram shown in Fig 5B, where the PC1 loadings were used as the input, closely followed the expected phylogenetic relationship based on what is known of the genetics and phylogenies of this complex [85,86]. *Compacta* and *japonica*, varieties of the same species [57], were paired together. Both of the species *Reynoutria japonica*, these plants are morphologically similar and often confused [17], consistent with the close linkage. The interbreeding species and hybrids were present in a different cluster to the five more distantly related species from within the Polygonaceae family. Although of the *Fallopia* genus, *Fallopia cilinodis* places separately to other *Fallopia* and *Reynoutria* species in a previous likelihood tree produced using a molecular dataset (nrITS, matK, trnL-trnF) [56]. *Fallopia cilinodis'* placement in a cluster separate to the interbreeding species is also consistent with molecular studies [56].

The placement of *Rumex acetosella* as the furthest from *Reynoutria japonica* var. *japonica* is unexpected. Molecular studies place *Rumex* closer than *Fagopyrum* [56]. Additionally, two potential biocontrol agents, the knotweed sawfly (*Allantus luctifer*) and a leaf beetle (*Gallerucida bifasciata* Motschulsky), were ruled out as candidates after host range testing confirmed it would feed on various native UK *Rumex* species [87]. This may be indicative of a similar composition of secondary metabolites to the target plant, Japanese Knotweed.

Importantly the hierarchical cluster analysis dendrograms in Fig 5 show that environmental factors can play a role in the determined linkages. Therefore, caution must be taken when using this method as a tool for phylogeny. Although the present study clearly demonstrates the power of this approach for the analysis of historic herbarium samples, fresh samples taken from plants grown together under controlled conditions prior to spectral acquisition would remove 'environment' as a variable to allow an unfettered comparison between plant types.

## Conclusion

We show ATR-FTIR spectroscopy coupled with SVM can accurately differentiate between leaf surfaces, plant types, and samples from different geographical locations even in herbarium samples of varying age of closely related species within the Polygonaceae family. This provides a rapid and robust method for hybrid identification, allowing informed decisions to be made regarding targeted control measures to tackle this invasive alien weed. This could be applied in the field using handheld mid and near-infra red devices [88,89]. Additionally, we found that spectra from the invasive *Reynoutria japonica* var. *japonica* knotweed variety indicated the presence of two polysaccharides, glucomannan and xyloglucan, in higher concentrations than in the dwarf variety. We have shown that ATR-FTIR spectroscopy and hierarchical cluster analysis provides an additional methodology for investigating linkage between closely related species. Adoption of this technology for the study of historic samples would increase the value of existing herbarium collections, which are currently threatened.

## Supporting information

**S1 Fig. Baseline corrected spectra.**
(PDF)

**S2 Fig.** (a) PCA scores plot, (b) LDA 2D scatter plot, (c) SVM scores plot and (d) SVM classification table of fingerprint spectra grouped by leaf surface: upper (blue) and lower (yellow) leaf surfaces.
(PDF)

**S3 Fig. PCA-LDA loadings graphs and key wavenumbers used for HCA analysis.**
(PDF)

**S4 Fig.** Scanning Electron Microscope images of the lower epidermis of **(a)** *Reynoutira japonica* and **(b)** *Reynoutria sachalinensis*.
(PDF)

**S5 Fig. HCA analysis using fingerprint region for each sample with species and location information.** See attached MATLAB file to zoom.
(PDF)

**S6 Fig. A zoomable MATLAB figure file of S5 Fig showing the HCA analysis dendrogram using the fingerprint region for each sample, with species and location information.**
(FIG)

**S1 Table. Herbarium sample information.**
(PDF)

**S2 Table. Quality parameters (accuracy, sensitivity, and specificity) for spectral classification of *Reynoutria japonica* var. *japonica* based on geographical location by PCA-LDA.**
(PDF)

**S3 Table. Quality parameters (accuracy, sensitivity, and specificity) for spectral classification based on sample type of closely related species, hybrids, and varieties by PCA-LDA.**
(PDF)

**S4 Table. Quality parameters for spectral classification based on sample type of closely related species, hybrids, and varieties by SVM.**
(PDF)

**S5 Table. Quality parameters (accuracy, sensitivity, and specificity) for spectral classification of *R. japonica* var. *japonica* based on geographical location by PCA-LDA.**
(PDF)

**S1 File. Spreadsheet containing the raw ATR-FTIR spectral data absorbances.**
(XLSX)

## Acknowledgments

The authors would like to thank the contributors to University of Leicester herbarium (LTR) for supplying the samples analysed in this study.

## Author Contributions

**Conceptualization:** Claire Anne Holden.

**Formal analysis:** Claire Anne Holden.

**Investigation:** Claire Anne Holden.

**Methodology:** Claire Anne Holden.

**Resources:** John Paul Bailey, Frank Martin.

**Supervision:** Jane Elizabeth Taylor, Martin McAinsh.

**Validation:** Paul Beckett.

**Writing – original draft:** Claire Anne Holden, Martin McAinsh.

**Writing – review & editing:** Claire Anne Holden.

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
