## [Decision Letter · Decision Letter 0]

26 Nov 2021

PONE-D-21-33488Know your enemy: Application of ATR-FTIR spectroscopy to invasive species controlPLOS ONE

Dear Dr. Holden,

Thank you for submitting your manuscript to PLOS ONE. After careful consideration, we feel that it has merit but does not fully meet PLOS ONE’s publication criteria as it currently stands. Therefore, we invite you to submit a revised version of the manuscript that addresses the points raised during the review process.

We look forward to receiving your revised manuscript.

Kind regards,

Du Changwen

Academic Editor

PLOS ONE

Journal Requirements:

"CAH is a member of the Centre for Global Eco-Innovation that is funded by the European Union Regional Development Fund and mediates the collaboration between Lancaster University and Phlorum Ltd. PB, of Phlorum Ltd, provided funding for CAH's studentship and expertise in Japanese Knotweed. " 

We note that you received funding from a commercial source: Phlorum Ltd

"CAH is a member of the Centre for Global Eco-Innovation that is funded by the European Union Regional Development Fund and mediates the collaboration between Lancaster University and Phlorum Ltd. The authors declare that there is no conflict of interest."

"CAH is a member of the Centre for Global Eco-Innovation that is funded by the European Union Regional Development Fund and mediates the collaboration between Lancaster University and Phlorum Ltd. PB, of Phlorum Ltd, provided funding for CAH's studentship and expertise in Japanese Knotweed. "

Additional Editor Comments:

The method made sense for the species discrimination, and the Reviwere gave a positive conment, I totally agree, I have gone through the manuscript, and some concerns raised as follwing:

1. The leaf samples are fresh or dry? is there pretreatment?

2. How to keep consistent contact without pressure in the FTIR-ATR recording?

3. In the materials and methods section, the wavenuber range is 4-4000 cm-1, but Figure 1 showes the range of 900-1800cm-1, how to confirm that this range is sufficient? how about other range?

4. Figure ciations should include more details. Such as , Figure 1, waht is the algorithm of spectra pre-processing? wavenumber range should be noted in Figure 2 and Figure 3.

5. Language problems should be considered, and language expression are sugested to be thorough checked and revised by native english speaker.

Reviewers' comments:

Reviewer's Responses to Questions

**Comments to the Author**

1. Is the manuscript technically sound, and do the data support the conclusions?

Reviewer #1: Yes

2. Has the statistical analysis been performed appropriately and rigorously? 

Reviewer #1: Yes

3. Have the authors made all data underlying the findings in their manuscript fully available?

Reviewer #1: Yes

4. Is the manuscript presented in an intelligible fashion and written in standard English?

Reviewer #1: Yes

5. Review Comments to the Author

Reviewer #1: The manuscript entitled "Know your enemy: Application of ATR-FTIR spectroscopy to invasive species control" addresses a relevant topic, that of the rapid identification of different species of invasive herbs, using the technique of vibrational FTIR spectroscopy. I am of the opinion that it should be accepted for publication after introducing the suggested corrections below.

Corrections needed:

line 19 - The only taxonomic categories that can be written in italics are genus, species, and categories below them.

line 22 - Indicate the species name (note: Indicating the name of varieties without first indicating the name of the species is not taxonomically correct!)

line 23 - Indicate the species name (note: Indicating the name of varieties without first indicating the name of the species is not taxonomically correct!)

line 24 - The only taxonomic categories that can be written in italics are genus, species, and categories below them.

line 28 - The same as line 22 and 23

line 60 - In each paragraph, the authors must always indicate the extensive and complete form of the names of the plants studied, so they should not limit themselves to indicating the initial letter "R."

line 65 - The abbreviation "var." must not be written in italics

line 277 - "compacta" and "japonica" in italics, please

line 297 - The only taxonomic categories that can be written in italics are genus, species, and categories below them.

line 306 - "compacta" and "japonica" in italics, please

line 308 - The same as in line 306

line 321 - The same as in line 306

line 416 - The abbreviation "var." must not be written in italics

line 429 - The only taxonomic categories that can be written in italics are genus, species, and categories below them.

line 436 - "Motschulsky" is the name of the species authority, so is not written in italics

line 449 - The only taxonomic categories that can be written in italics are genus, species, and categories below them.

6. PLOS authors have the option to publish the peer review history of their article (what does this mean?). If published, this will include your full peer review and any attached files.

Reviewer #1: **Yes: **Leonel Pereira

---

## [Author Response · Author response to Decision Letter 0]

2 Dec 2021

Response to Reviewers

We thank Professor Leonel Pereira and Professor Du Changwen for their positive and constructive response to our manuscript and for their valuable comments as to how the submission can be improved still further. We welcome the opportunity to respond to each of the comments below. Changes made to the manuscript in response to the reviewer’s comments are tracked in word in the revised submission.

Reviewer One - Professor Leonel Pereira

line 19 - The only taxonomic categories that can be written in italics are genus, species, and categories below them.

Italics have been removed from ‘Polygonaceae’.

line 22 - Indicate the species name (note: Indicating the name of varieties without first indicating the name of the species is not taxonomically correct!)

‘Reynoutria’ has been added.

line 23 - Indicate the species name (note: Indicating the name of varieties without first indicating the name of the species is not taxonomically correct!)

‘Reynoutria’ has been added.

line 24 - The only taxonomic categories that can be written in italics are genus, species, and categories below them.

Italics have been removed from ‘Polygonaceae’.

line 28 - The same as line 22 and 23

‘Reynoutria’ has been added to both mentions.

line 60 - In each paragraph, the authors must always indicate the extensive and complete form of the names of the plants studied, so they should not limit themselves to indicating the initial letter "R."

We have expanded all abbreviations of R. to Reynoutria, and F. to Fallopia.

line 65 - The abbreviation "var." must not be written in italics

Italics have been removed from ‘var’.

line 277 - "compacta" and "japonica" in italics, please

Italics have been added.

line 297 - The only taxonomic categories that can be written in italics are genus, species, and categories below them.

Italics have been removed from ‘Polygonaceae’.

line 306 - "compacta" and "japonica" in italics, please

Italics have been added.

line 308 - The same as in line 306

Italics have been added.

line 321 - The same as in line 306

Italics have been added.

line 416 - The abbreviation "var." must not be written in italics

Italics have been removed from ‘var’.

line 429 - The only taxonomic categories that can be written in italics are genus, species, and categories below them.

Italics have been removed from ‘Polygonaceae’.

line 436 - "Motschulsky" is the name of the species authority, so is not written in italics

Italics have been removed from ‘Motschulsky’.

line 449 - The only taxonomic categories that can be written in italics are genus, species, and categories below them.

Italics have been removed from ‘Polygonaceae’.

Additional Editor Comments - Professor Du Changwen

‘The leaf samples are fresh or dry? is there pre-treatment?’

The following sentence has been added to the materials and methods, line 120:

‘Leaves were air-dried at the time of collection and subsequently stored in cardboard folders within purpose-built cupboards to protect them from light exposure.’

‘How to keep consistent contact without pressure in the FTIR-ATR recording?’

Consistent contact was maintained by raising the leaf material mounted on a slide to meet the internal reflection element using a moving platform, as described in lines 138-140.

‘In the materials and methods section, the wavenumber range is 4-4000 cm-1, but Figure 1 shows the range of 900-1800 cm-1, how to confirm that this range is sufficient? how about other range?’

An explanation of this choice and an acknowledgement of other available options has been added to the discussion section, lines 403-412.

‘In the present study, the fingerprint region (1800–900 cm− 1) of acquired spectra was selected for chemometric analysis because biological molecules preferentially absorb light of these wavenumbers, including important biological absorptions due to lipids, proteins, carbohydrates, nucleic acids and protein phosphorylation [26]. Isolation of this fingerprint region (1800–900 cm− 1) has achieved good results in other plant studies [29,39,40,73–75], though the high region (3700-2800 cm–1) has also yielded valuable information in a range of applications [76–79] since it contains additional biologically relevant absorbances such as those for water (~3275 cm–1), protein (~3132 cm-1), fatty acids and lipids (~3005, ~2970, ~2942 and ~2855 cm–1) [26].

‘Figure citations should include more details. Such as, Figure 1, what is the algorithm of spectra pre-processing? wavenumber range should be noted in Figure 2 and Figure 3.’

The wavenumber range for the fingerprint region ‘(1800-900 cm-1)’ has been added in brackets in figure captions for Figures 2, 3, and 5. The pre-processing used has been added to all figure captions.

‘Language problems should be considered, and language expression are suggested to be thorough checked and revised by native English speaker.’

The manuscript has been checked by a native English speaker and we believe it meets the high standards expected of a PLOS ONE publication.

Journal Requirements

Reference list

The reference list is complete and correct.

Stylisation

Level one heading front has been made bold and 18 pt. Level two heading font has been changed to bold and 16 pt. Referral to ‘Figure’ has been shortened to ‘Fig’. ‘Supplementary Figure S1’ has been changed to ‘S1 Fig’. Supporting Information has been listed after the references. The file names have been changed to adhere to the PLOS ONE author guidelines. Author affiliations have been changed to a numerical system. The corresponding author has been designated with an asterisk.

Competing Interests

We have amended the cover letter to outline that the funding was received from a commercial source and can confirm that this does not alter our adherence to PLOS ONE policies on sharing data and materials.

Funding

We have amended the cover letter to outline that the funding received from a commercial source and removed mention of funders from the acknowledgement section within the manuscript.

---

## [Editor Report · Decision Letter 1]

9 Dec 2021

Know your enemy: Application of ATR-FTIR spectroscopy to invasive species control

PONE-D-21-33488R1

Dear Dr. Holden,

We’re pleased to inform you that your manuscript has been judged scientifically suitable for publication and will be formally accepted for publication once it meets all outstanding technical requirements.

Kind regards,

Du Changwen

Academic Editor

PLOS ONE

Additional Editor Comments (optional):

The revisions are satifactory, and the manuscript is ready for publication.
---

## [Editor Report · Acceptance letter]

27 Dec 2021

PONE-D-21-33488R1 

Know your enemy: Application of ATR-FTIR spectroscopy to invasive species control 

Dear Dr. Holden:

I'm pleased to inform you that your manuscript has been deemed suitable for publication in PLOS ONE. Congratulations! Your manuscript is now with our production department. 

Kind regards, 

on behalf of

Professor Du Changwen 

Academic Editor

PLOS ONE